# Alkenone isotopes show evidence of active carbon concentrating mechanisms in coccolithophores as aqueous carbon dioxide concentrations fall below 7 $\mathrm{\mu mol L^{-1}}$

Marcus P. S. Badger[1]

[1]School of Environment, Earth & Ecosystem Sciences, The Open University, Milton Keynes, MK7 6AA, UK

**Correspondence:** Marcus P. S. Badger (marcus.badger@open.ac.uk)

**Abstract.**

Coccolithophores and other haptophyte algae acquire the carbon required for metabolic processes from the water in which they live. Whether carbon is actively moved across the cell membrane via a carbon concentrating mechanism, or passively through diffusion, is important for haptophyte biochemistry. The possible utilisation of carbon concentrating mechanisms also

has the potential to over-print one proxy method by which ancient atmospheric $CO_2$ concentration is reconstructed using alkenone isotopes. Here I show that carbon concentrating mechanisms are likely used when aqueous carbon dioxide concentrations are below 7 µmol L⁻¹. I compile published alkenone based $CO_2$ reconstructions from multiple sites over the Pleistocene and recalculate them using a common methodology, which allows comparison to be made with ice core $CO_2$ records. Interrogating these records reveal that the relationship between proxy- and ice core- $CO_2$ breaks down when local aqueous $CO_2$

concentration falls below 7 µmol L⁻¹. The recognition of this threshold explains why many alkenone based $CO_2$ records fail to accurately replicate ice core $CO_2$ records, and suggests the alkenone proxy is likely robust for much of the Cenozoic when this threshold was unlikely to be reached in much of the global ocean.

## 1   Introduction

Alkenones are long–chain ($C_{37-39}$) ethyl– and methy– ketones (Figure 1; Brassell et al. (1986); Rechka and Maxwell (1987)) produced by a restricted group of photosynthetic haptophyte algae (Conte et al., 1994). Produced by a narrow group of organisms which live exclusively in the photic zone, alkenones allow probing of algal biogeochemistry, and as alkenones are often preserved in the sedimentary record, alkenones can also provide information about past environmental conditions.

    Two main proxy systems based on alkenone geochemistry exist, one allows reconstruction of sea surface temperature (SST)

and relies on the changing degree of unsaturation of the $C_{37}$ alkenone ($U_{37}^{K'}$) (Brassell et al., 1986) whilst a second for atmospheric $CO_2$ concentration is based on reconstructing the isotopic fractionation which takes place during photosynthesis ($\varepsilon_\mathrm{p}$) (Laws et al., 1995; Bidigare et al., 1997). It is the second system using the stable carbon isotopic composition of the preserved

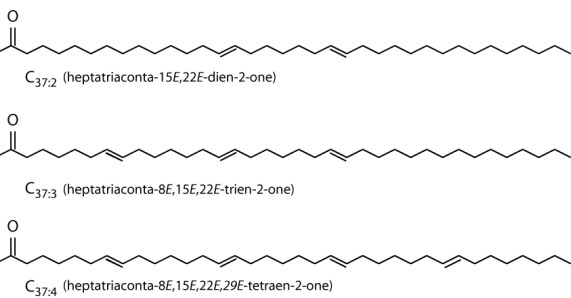

**Figure 1.** Alkenones are $C_{37}$ unsaturated methyl ketones (Brassell et al., 1986; Rechka and Maxwell, 1987).

alkenones for reconstructing atmospheric $CO_2$ concentration (referred to throughout as $CO_{2(\varepsilon_p - alk)}$) which is the focus of this study.

In the modern ocean alkenones are produced primarily by two dominant coccolithophore species; *Emiliania huxleyi* and *Gephyrocapsa oceanica*. *E. huxleyi* first appeared 290 kyr ago, and began to dominate over *G. oceanica* around 82 kyrs ago (Gradstein et al., 2012; Raffi et al., 2006). However alkenones are commonly found in sediments throughout the Cenozoic, with the oldest reported detections from mid-Albian aged black shales (Farrimond et al., 1986). Prior to the evolution of *G. oceanica* alkenones were most likely produced by other closely related species from the Noelaerhabdaceae family (Marlowe

et al., 1990; Volkman, 2000). Micropaleontological and molecular data split the coccolith-bearing haptophytes into two distinct phylogenetic clades; the Isochrysidales and Coccolithales. The Isochrysidales contain the modern alkenone producing taxa including *E. huxleyi* and *G. oceanica*, and fossil reticulofenestrids. Meanwhile the non-alkeone producers are separated into the order Coccolithales which includes *Coccolithus pelagicus* and *Calcidiscus leptoporus* along with most other coccolithophores.

Proxies for atmospheric $CO_2$ concentration including $CO_{2(\varepsilon_p - alk)}$, those based on the $\delta^{11}B$ of planktic foraminifera, geo-

chemical modelling and stomatal density, broadly agree that over the Cenozoic atmospheric $pCO_2$ declined from high levels ( >1000 µatm) in the "greenhouse" worlds of the Paleocene and Eocene to close to modern day values (around 400 µatm) in the Pliocene (Pagani et al., 2005, 2011; Pearson et al., 2009; Anagnostou et al., 2016; Foster et al., 2017; Sosdian et al., 2018; Super et al., 2018; Zhang et al., 2013; Beerling and Royer, 2011). However, recently discrepencies have emerged between $CO_{2(\varepsilon_p - alk)}$ and other $CO_2$ proxies at the <400 µatm atmospheric $CO_2$ concentrations of the Pleistocene (i.e. Badger et al.

(2019, 2013a) and compare Badger et al. (2013b) and Pagani et al. (2009) with Martínez-Botí et al. (2015)). Whilst the long-standing differences between alkenone (Pagani et al., 1999), $\delta^{11}B$ (Foster et al., 2012) and stomatal proxies (Kürschner et al., 2008) in the Miocene $CO_2$ reconstructions have been partially resolved with new SST records (Super et al., 2018), differences remain in the Pliocene (Pagani et al., 2009; Badger et al., 2013b; Martínez-Botí et al., 2015) and Pleistocene (Badger et al., 2019).

## 1.1 Carbon Concentrating mechanisms

One plausible reason for the discrepancies between $CO_{2(\varepsilon_p-alk)}$ and other proxies for atmospheric $CO_2$ is the operation of active carbon concentrating mechanisms (CCMs) in haptophytes. These are potentially important as $CO_{2(\varepsilon_p-alk)}$ assumes purely passive uptake of carbon into the haptophyte cell purely via diffusion (Laws et al., 1995; Bidigare et al., 1997). The potential for CCMs to effect $CO_{2(\varepsilon_p-alk)}$ has long been known (Laws et al., 1997, 2002; Cassar et al., 2006) and recent work has refocussed efforts on understanding CCMs in $CO_{2(\varepsilon_p-alk)}$ (Bolton et al., 2012; Bolton and Stoll, 2013; Stoll et al., 2019; Zhang et al., 2019, 2020). Coccolithophores are thought to have low efficiency CCMs, especially compared to diatoms, dinoflagellates and *Phaeocystis*, with evidence that CCMs play a minor role in coccolithophore biochemistry in the $CO_2$ replete worlds of the early Cenozoic (Bolton et al., 2012; Reinfelder, 2011). Direct evidence from experimentation with the marine diatom *Phaeodactylum tricornutum* suggest that both passive diffusive uptake and active CCMs operate at the same time, with active uptake used to moderate internal cell $CO_2$ concentrations to minimise energy use during transport to carboxylation sites (Laws et al., 1997). $CO_2$, unlike some other nutrients, is replete within the water column, especially when considering the DIC reservoir which includes bicarbonate ($HCO_3^-$), carbonate ($CO_3^{2-}$) and dissolved $CO_2$ ($[CO_2]_{(aq)}$) . However, due to the relatively slow diffusion of dissolved $[CO_2]_{(aq)}$ through water and the slow kinetics of the bicarbonate to $[CO_2]_{(aq)}$ transformation, surface water $[CO_2]_{(aq)}$ can still be depleted by photosynthetic activity. This can become particularly problematic in species which form blooms, and at the cell boundary of species with limited motility. It should be no surprise therefore that many marine photosynthetic organisms have evolved with mechanisms to concentrate carbon within the cell.

The enzyme carbonic anhydrase (CA) can catalyse the dehydration of $HCO_3^-$ to $[CO_2]_{(aq)}$ to speed up availability of carbon if the $[CO_2]_{(aq)}$ reservoir is depleted and has been observed in several haptophytes including coccolithophores (Rost et al., 2003; Riebesell et al., 2007). The exact contribution of CA remains unclear but two possible mechanisms for CCMs have been postulated (Reinfelder, 2011) (1) CA catalyzes dehydration of $HCO_3^-$ at the cell surface, which then allows increased $CO_2$ to diffuse into the cell passively and (2) $HCO_3^-$ is transported into the cell and then converted by CA. Both of these options will likely impact the $CO_{2(\varepsilon_p-alk)}$ proxy, firstly by changing the effective $[CO_2]_{(aq)}$ within the cell (and so impacting $\varepsilon_p$), and secondly by imparting another carbon isotopic fractionation during CA catalyzation which is not considered by the $CO_{2(\varepsilon_p-alk)}$ proxy system. However CA activity in coccolithophores does not appear to be regulated by $CO_2$ as it is in diatoms and *Phaeocystis* (Rost et al., 2003) which may indicate a less well developed CCM in coccolithophores.

Calcifying coccolithophores (which include alkenone producers *E. huxleyi* and *G. oceanica*) may be able to utilize $HCO_3^-$ directly as a carbon source (Trimborn et al., 2007), with precipitation of $CaCO_3$ providing an acid for the dehydration of $HCO_3^-$ but this still requires sufficent $HCO_3^-$ entering the cell and it is unclear whether calcification aids DIC acquisition (Riebesell et al., 2000; Zondervan et al., 2002). The light dependant leak of carbon (as $CO_2$ and DIC) back from haptophyte cells (including the coccolithophore *E. huxleyi*) to seawater (Tchernov et al., 2003) suggests that CCMs are energy intensive and can concentrate DIC within the cell. Even with active CCMs, it appears that in the ocean coccolithophores are $CO_2$ limited under some circumstances (Riebesell et al., 2007).

**Table 1.** Sites with Pleistocene $CO_{2(\varepsilon_p-alk)}$ records. Note that the MANOP Site C record was generated to track changes in surface water–atmosphere equilibrium not atmospheric $pCO_2$, so although included here for completeness, is not included in the analysis. Distance from coast is calculated from the intermediate resolution version of GSHHG and computed using Generic Mapping Tools (GMT) (Wessel and Smith, 1996; Wessel et al., 2019)

| Site | Age interval (kyr) | Latitude | Longitude | Water depth (m) | Distance from coast (km) | Reference |
|------|-----|-----|-----|-----|-----|-----|
| 05PC-21 | 0.5–188 | 38° 24'N | 131° 33'E | 1721 | 108 | Bae et al. (2015) |
| DSDP 619 | 3–92 | 27° 11.61' N | 91° 24.54' W | 2259 | 489 | Jasper and Hayes (1990) |
| NIOP 464 | 7.8–29 | 22° 9' N | 63° 21'E | 1470 | 333 | Palmer et al. (2010) |
| ODP 999 | 111–258 | 12° 44.639' N | 78° 44.360' W | 2839 | 249 | Badger et al. (2019) |
| ODP 925 | 20–580 | 4° 12.249' N | 43° 29.334' W | 3042 | 626 | Zhang et al. (2013) |
| MANOP Site C | 0.8–253 | 0° 57.2" N | 138° 57.3' W | 4287 | 998 | Jasper et al. (1994) |
| GeoB 1016-3 | 1.3–196 | 11° 46.2' S | 11° 40.9' E | 3410 | 185 | Andersen et al. (1999) |

*The full record for ODP Site 925 extends to 38.62 Ma

## 2 Materials and Methods

### 2.1 Calculating $CO_2$ from alkenone $\delta^{13}C$ values: The $CO_{2(\varepsilon_p-alk)}$ proxy

In this study I use the now large number of published $CO_{2(\varepsilon_p-alk)}$ records which overlap with ice core records of atmospheric $CO_2$ concentration (Tables 1 & 2) to explore the relationship between $CO_{2(\varepsilon_p-alk)}$ and CCMs in the Pleistocene, where our understanding of atmospheric $CO_2$ concentration is best.

Multiple records of $CO_{2(\varepsilon_p-alk)}$ have been published for the Pleistocene (Figure 2, Table 1) allowing direct comparision with ice core based $CO_2$ records (Table 2). These records are globally distributed in longitude, but are concentrated at low lati-
tude sites, largely as there is a general preference for sites which have (in the modern ocean) surface waters close to equilibrium with the atmosphere (Figure 2, Table 1). In longer term palaeoclimate studies there has also been a preference for low latitude, gyre sites in the belief that these sites are more likely to be oceanographically stable over long time intervals (Pagani et al., 1999). Most of the records included here (Table 1, Figure 2) were generated with the aim to reconstruct atmospheric $CO_2$ concentration, however one, the MANOP C Site of Jasper et al. (1994), was used to explicitly reconstruct changing disequilibrium
due to oceanographic frontal changes over time, and so is excluded from the following analysis.

Whilst these sites do only span a relatively small latitudinal extent, the diversity of settings does allow for investigation of any secondary controls on alkenone $\delta^{13}C$ values ($\delta^{13}C_{alkenone}$). In particular, differences in oceanographic setting and SST to test the hypothesis that low $[CO_2]_{(aq)}$ breaks the relationship between $\delta^{13}C_{alkenone}$ and atmospheric $CO_2$ concentration, as might be expected if haptophytes are able to actively take up carbon from seawater to meet metabolic demand – i.e. activate
CCMs.

**Table 2.** Sources of ice core data, as compiled by Bereiter et al. (2015). WAIS - West Antarctic Ice Sheet, TALDICE - TALos Dome Ice CorE, EDML - EPICA Dronning Maud Land

| Age interval (kyr) | Ice core location | Reference |
| --- | --- | --- |
| −0.051–1.8 | Law Dome | Rubino et al. (2013) |
| 1.8–2 | Law Dome | MacFarling Meure et al. (2006) |
| 2–11 | Dome C | Monnin et al. (2001, 2004) |
| 11–22 | WAIS | Marcott et al. (2014) |
| 22–40 | Siple Dome | Ahn and Brook (2014) |
| 40–60 | TALDICE | Bereiter et al. (2012) |
| 60–115 | EDML | Bereiter et al. (2012) |
| 105–155 | Dome C Sublimation | Schneider et al. (2013) |
| 155–393 | Vostok | Petit et al. (1999) |

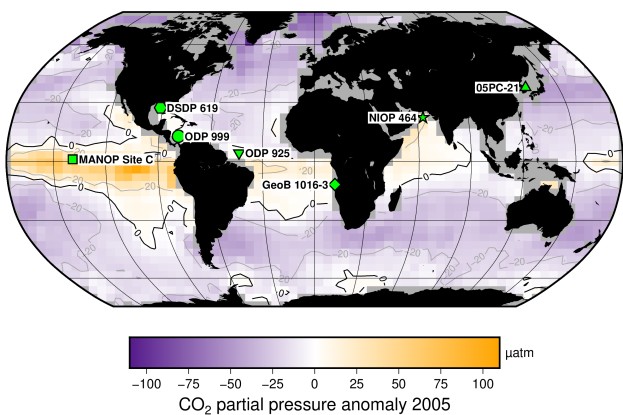

**Figure 2.** Study sites relative to mean annual surface ocean $CO_2$ disequilibrium for 2005. Sites are globally distributed in longitude but restricted in latitude, as generally sites are chosen to be close to surface water equilibrium with the atmosphere. Sites used for this study are indicated, over the mean annual surface ocean disequilibrium for 2005 calculated from Takahashi et al. (2014). The MANOP C Site (Jasper et al., 1994) was choosen to study the disequilbrium at that site, so is shown here but not used in the following analyses. Site symbols are used throughout the figures: ODP 999 - circle, 05PC-21 - triangle, ODP 925 - inverted triangle, DSDP 619 - hexagon, MANOP Site C - square, NIOP 464 - star, GeoB 1016-3 - diamond.

To facilitate fair comparision between sites and consistent comparision with the ice core records, all $CO_{2(\varepsilon_p - \text{alk})}$ records were recalculated using a consistent approach. The approach is based on Bidigare et al. (1997) which updated the initial approach of Jasper and Hayes (1990) to $CO_{2(\varepsilon_p - \text{alk})}$. This approach removes some additional corrections used in the original publication of the records (such as growth rate adjustment for NIOP 464 (Palmer et al., 2010)) but does allow for direct

comparison to be made. For all sites the 'b' term was estimated using modern day surface $[PO_4^{3-}]$ (Bidigare et al., 1997; Pagani et al., 2009)

An overview of how $CO_{2(\varepsilon_p-alk)}$ data are typically generated is given in Badger et al. (2013b). Briefly, to calculate $\varepsilon_p$ requires the stable carbon isotopic composition of the dissolved $CO_2$ ($\delta^{13}C_{CO_{2(aq)}}$) and haptophyte biomass ($\delta^{13}C_{org}$). The isotopic fractionation between $\delta^{13}C_{alkenone}$ and $\delta^{13}C_{org}$ is first corrected assuming a constant fractionation ($\varepsilon_{alkenone}$) of 4.2 ‰ (Garcia et al., 2013; Popp et al., 1998; Bidigare et al., 1997):

$$\varepsilon_{alkenone} = \frac{\delta^{13}C_{alkenone} + 1000}{\delta^{13}C_{org} + 1000} - 1 \tag{1}$$

The isotopic composition of dissolved inorganic carbon (DIC) is estimated using (ideally) the $\delta^{13}C$ value of planktic foraminifera and the temperature-dependant fractionation between calcite and $[CO_2]_{(g)}$ experimentally determined by Romanek et al. (1992), where T is sea surface temperature in degrees Celsius (SST):

$$\varepsilon_{calcite-CO_{2(g)}} = 11.98 - 0.12T \tag{2}$$

The value the carbon isotopic composition of $CO_{2(g)}$ ($\delta^{13}C_{CO_{2(g)}}$) can then be calculated:

$$\delta^{13}C_{CO_{2(g)}} = \frac{\delta^{13}C_{carbonate} + 1000}{\varepsilon_{calcite-CO_{2(g)}}/1000 + 1} - 1000 \tag{3}$$

From this $\delta^{13}C_{CO_{2(aq)}}$ can be calculated using the relationship experimentally determined by Mook et al. (1974):

$$\varepsilon_{CO_{2(aq)}-CO_{2(g)}} = \frac{-373}{T + 273.15} + 0.19 \tag{4}$$

and

$$\delta^{13}C_{CO_{2(aq)}} = \left(\frac{\varepsilon_{CO_{2(aq)}-CO_{2(g)}}}{1000} + 1\right)$$
$$\cdot \left(\delta^{13}C_{CO_{c(g)}} + 1000\right)$$
$$- 1000 \tag{5}$$

Finally $\varepsilon_p$ can be calculated:

$$\varepsilon_p = \left(\frac{\delta^{13}C_{CO_{2(aq)}} + 1000}{\delta^{13}C_{org} + 1000} - 1\right) \cdot 1000 \tag{6}$$

and from that $[CO_2]_{(aq)}$ is calculated using the isotopic fractionation during carbon fixation ($\varepsilon_f$) and 'b', which represents the summation of physiological factors:

$$[CO_2]_{(aq)} = \frac{b}{\varepsilon_f - \varepsilon_p} \tag{7}$$

Here $\varepsilon_f$ is assumed to be a constant 25 ‰ (Bidigare et al., 1997). In the modern ocean the 'b' term, which accounts for physiological factors such as cell size and growth rate, shows a close correlation with $[PO_4^{3-}]$ (Bidigare et al., 1997; Pagani et al., 2009). However, the relationship between $b$, growth rate and $[PO_4^{3-}]$ has recently been questioned (Zhang et al., 2019, 2020) but for the purposes of this analysis is assumed to hold. This is discussed further below. Values for SST, $\delta^{13}C_{alkenone}$, $\delta^{13}C_{carbonate}$, salinity and $[PO_4^{3-}]$ are either taken from the original publications or estimated from modern ocean estimates (Takahashi et al., 2009; Antonov et al., 2010; Garcia et al., 2013; Locarnini et al., 2013).

Providing that the atmosphere is in equilibrium with surface water, the concentration of atmospheric $CO_2$ can be calculated from $[CO_2]_{(aq)}$, (and vice versa if atmospheric $CO_2$ concentration is known) using Henry's law:

$$pCO_2 = \frac{[CO_2]_{(aq)}}{K_H} \tag{8}$$

The solubility coefficient ($K_H$) is dependant on salinity and SST, and here is calculated following the parameterization of Weiss (1970, 1974).

## 3 Results

### 3.1 Multi-site comparisons between $CO_{2(\varepsilon_p-alk)}$ and the ice core records

Across the six sites included in this analysis, there are 217 $CO_{2(\varepsilon_p-alk)}$-based estimates of atmospheric $CO_2$ concentration over the past 260 Ka for comparison with the ice core records (Table 2; Bereiter et al. (2015)). When all $CO_{2(\varepsilon_p-alk)}$ estimates are considered together over 260 Ka, this compilation of proxy-based records fails to replicate the ice core record (Figure 3). This has already been noted at specific sites (e.g. Site 999 in the Caribbean Badger et al. (2019)) but this is the first time that all available records coincident with the Pleistocene ice core records have been compiled using a common methodology. Notably the $CO_{2(\varepsilon_p-alk)}$ based estimates are rarely lower than time-equivalent ice core estimate, but frequently higher. Given that haptophytes require carbon to satisfy metabolic demand, this is perhaps unsurprising; if at times of low carbon availability haptophytes can switch from passive to active uptake to satisfy metabolic demand, it would be times of low atmospheric $CO_2$ concentration (and so lower $[CO_2]_{(aq)}$) when the active uptake is most likely to be needed. As $CO_{2(\varepsilon_p-alk)}$-based estimates of atmospheric $CO_2$ concentration rely on the assumption of a purely diffusive uptake of carbon, it is therefore likely that the proxy would perform worse at times of low atmospheric $CO_2$ concentration.

The haptophytes do not directly interact with the atmosphere, obtaining their carbon from dissolved carbon. As it is not only atmospheric $CO_2$ concentration which controls the concentration of dissolved carbon ($[CO_2]_{(aq)}$), but also temperature, alkalinity and other oceanographic factors which control the equilibrium state between surface waters at the atmosphere, (Figure 2) the multiple sites in different settings now give the opportunity to test whether other factors are important in controlling the accuracy of $CO_{2(\varepsilon_p-alk)}$.

To produce time-equivalent estimates of atmospheric $CO_2$ concentration for comparison with the ice core records, a simple linear interpolation of the Bereiter et al. (2015) compilation was initially used (Figure 4). This assumes that both the age model

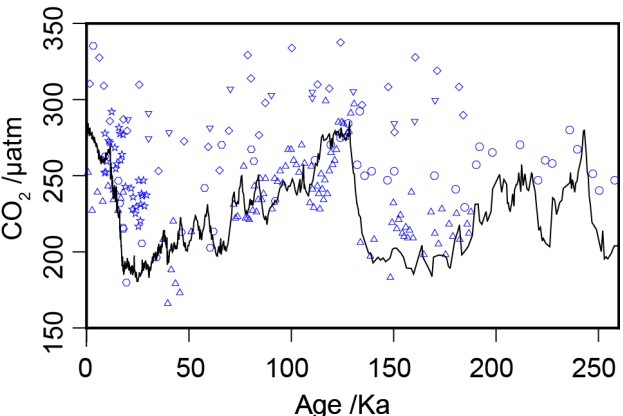

**Figure 3.** Compiled $CO_{2(\varepsilon_p - alk)}$-based estimates of atmospheric $CO_2$ concentration over the past 260 Ka (blue circles), with the ice core compilation of Bereiter et al. (2015) shown as the solid black line. Full sources for the ice core and $CO_{2(\varepsilon_p - alk)}$ records are in Tables 1 and 2.

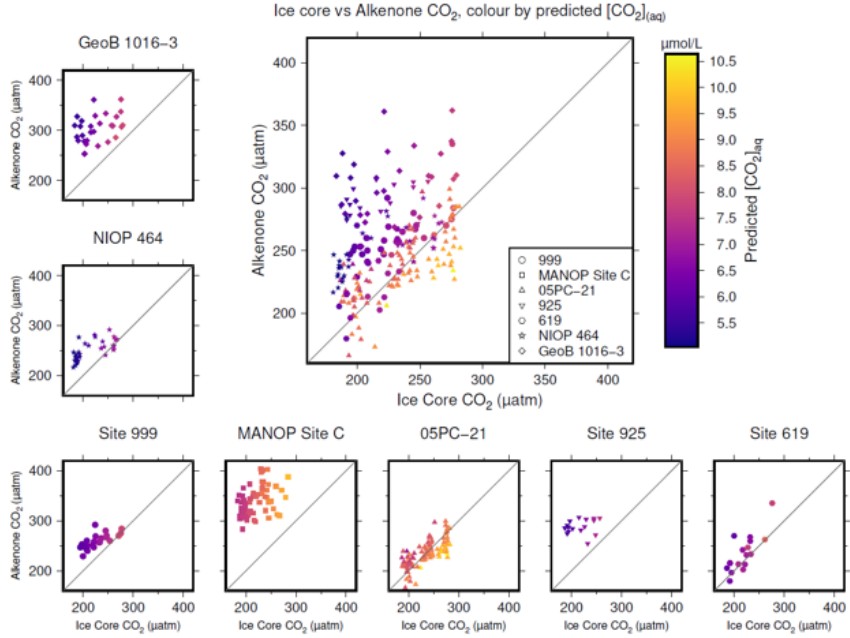

**Figure 4.** Crossplots of $CO_{2(\varepsilon_p - alk)}$-based atmospheric $CO_2$ concentration (y-axes) vs the time-equivalent estimate from ice core records (x-axes; Bereiter et al. (2015); Table 2)). The large panel compiles all sites, with the exception of MANOP Site C, as explained in the text. Symbols are coloured by predicted $[CO_2]_{(aq)}$ for each site and time as explained in the text. Full sources for alkenone data are shown in Table 1. A 1:1 line is included in all plots for comparison.

of the ice core and the published age models of the sites are correct and equivalent. This is almost certainly not the case, and so for the calculations below, a $\pm 3000$ year uncertainty is included for ages of both the ice core and $CO_{2(\varepsilon_p-\mathrm{alk})}$ values. Figure 4 shows that $CO_{2(\varepsilon_p-\mathrm{alk})}$-based atmospheric $CO_2$ concentration agree with ice core $CO_2$ at some sites and at some times, but not throughout. Sites 05-PC21 (Bae et al., 2015) and DSDP Site 619 (Jasper and Hayes, 1990) perform quite well, throughout, whilst ODP Site 999 (Badger et al., 2019) and NIOP 464 (Palmer et al., 2010) only appear to agree at higher values of $CO_2$, at ODP Site 925 (Zhang et al., 2013) and GEoB 1016-3 (Andersen et al., 1999) there is very little overlap between the two methods of reconstructing atmospheric $CO_2$ concentration.

To explore whether $[CO_2]_{(aq)}$ is an imporant influence on $CO_{2(\varepsilon_p-\mathrm{alk})}$, I calculate predicted $[CO_2]_{(aq)}$ ($[CO_2]_{(aq)-\mathrm{predicted}}$) for each of the samples. To calculate $[CO_2]_{(aq)-\mathrm{predicted}}$, the time-equivalent value of atmospheric $CO_2$ concentration from the ice core record is used in combination with Eq. 8 to calculate $[CO_2]_{(aq)}$ at the time of alkenone production for each sample. Reconstructed estimates of SST and salinity are used as for $CO_{2(\varepsilon_p-\mathrm{alk})}$ above, along with any estimated surface water-atmosphere disequilbrium. Points in Figure 4 are then coloured by $[CO_2]_{(aq)-\mathrm{predicted}}$.

Inspection of Figure 4 suggests a connection between ($[CO_2]_{(aq)-\mathrm{predicted}}$) and the skill of $CO_{2(\varepsilon_p-\mathrm{alk})}$ to reconstruct atmospheric $CO_2$ concentration. The points clustering around the 1:1 line are lighter in colour (so with higher $[CO_2]_{(aq)-\mathrm{predicted}}$), whilst points falling away from the 1:1 line have lower $[CO_2]_{(aq)-\mathrm{predicted}}$. To explore this relationship, I progressively restricted the included samples on the basis of $[CO_2]_{(aq)-\mathrm{predicted}}$, and at each stage calculated a Pearson correlation co-efficient ($r$) and coefficient of determination ($r^2$) for each subset. Under this analysis the correlation progressively increased as more of the low $[CO_2]_{(aq)-\mathrm{predicted}}$ samples were excluded (Figure 5). All analyses were performed in R (R Core Team, 2020) using RStudio (RStudio Team, 2020). This suggests that the fidelity of the $CO_{2(\varepsilon_p-\mathrm{alk})}$ depends on the concentration of $[CO_2]_{(aq)}$, improving at higher levels of $[CO_2]_{(aq)}$.

To further investigate this potential relationship, I progressively exclude samples based on $[CO_2]_{(aq)-\mathrm{predicted}}$ with a step size of 0.05 $\mu molL^{-1}$, again calculating Pearson correlation coefficients and coefficients of determination between ice core and $CO_{2(\varepsilon_p-\mathrm{alk})}$ for each subsample of the population. The result is shown in Figure 6. Here the analysis shows, similar to Figure 5, that as the samples with lowest $[CO_2]_{(aq)-\mathrm{predicted}}$ are progressively removed, the correlation between ice core and $CO_{2(\varepsilon_p-\mathrm{alk})}$ increases. Furthermore, this continues only up until $[CO_2]_{(aq)-\mathrm{predicted}}$ reaches 7 $\mu molL^{-1}$. Above this, the coefficient of determination plateaus, until the subsample reaches such a small size that spurious correlations become important (Figure 6b).

## 3.2   Sensitivity and Uncertainty Tests

It is possible that the pattern seen in Figure 6b could emerge from a dataset shaped with increasing densisity surrounding the 1:1 correlation line without being driven by changes in $[CO_2]_{(aq)-\mathrm{predicted}}$. To explore this possiblity I ran a series of sensitivity experiments. In these, rather than reducing the sample by filtering by $[CO_2]_{(aq)-\mathrm{predicted}}$, the whole dataset (Table 1) was randomly ordered, and then stepwise subsampled. To make this equivalent to the $[CO_2]_{(aq)-\mathrm{predicted}}$ analysis above, I set the size of each subsample to be equal to each step in the original analysis. This produces a randomly selected, but same sized sub-sample such that the size of the subsample reduces in the same way as shown in Figure 6b). Pearson correlation

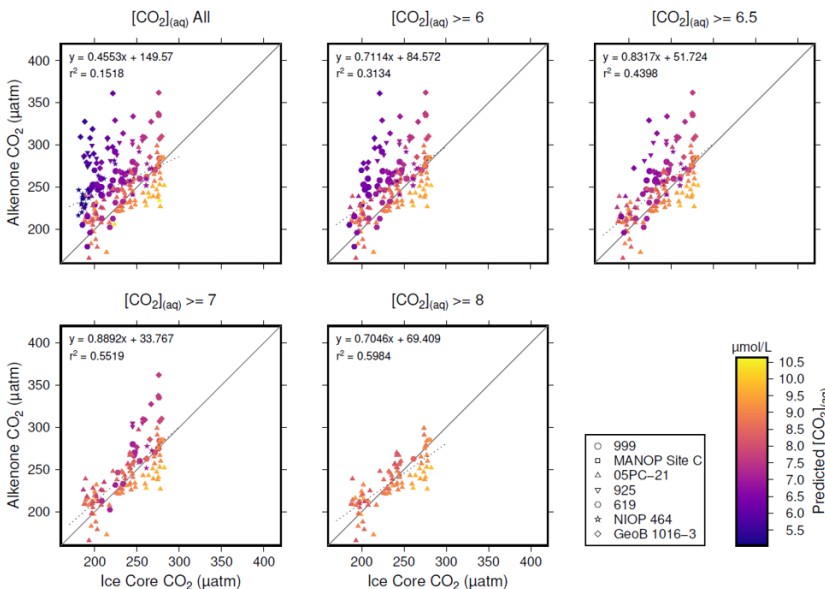

**Figure 5.** Crossplots of $CO_{2(\varepsilon_p-alk)}$-based atmospheric $CO_2$ concentration (Table 1; y-axes) vs the time-equivalent estimate from ice core records (x-axes; Bereiter et al. (2015); Table 2)). The sample of published vales of $CO_{2(\varepsilon_p-alk)}$ was progressively restricted by $[CO_2]_{(aq)-predicted}$, indicated by the subplot titles. Individual values are coloured by $[CO_2]_{(aq)-predicted}$, and Sites indicated by shape (see key). Coefficients of determination and equations of best fit are shown in each panel, along with a 1:1 line.

coefficients and coefficients of determination were calculated for each subsample as above, and I repeated this 1000 times, with the order of each sample randomized each time.

To allow for possible age model uncertainties, a 3000 year ($1\sigma$) uncertainty was also applied to each sample. This uncertainty was applied to the age of each sample prior to sampling of the ice core record, and is applied as a normally distributed uncertainty. Uncertainty in $CO_{2(\varepsilon_p-alk)}$ measurements is typically calculated using Monte Carlo modelling of all the parameters (i.e Pagani et al. (1999); Badger et al. (2013a, b)), however this was not done in all the published work (Table 1), and some differences in approach were found across the published work. Therefore to create $CO_{2(\varepsilon_p-alk)}$ uncertainty estimates for each value in this study, I emulate the uncertainties based on the $CO_{2(\varepsilon_p-alk)}$ value. I built a simple emulator (Figure 7) by running Monte Carlo uncertainty estimates for all of the included datasets (Table 1) using the same estimates of uncertainty for each variable in the $CO_{2(\varepsilon_p-alk)}$ calculation as applied in Badger et al. (2013a, b). This then allows the uncertainty to be included in the $[CO_2]_{(aq)-predicted}$ calcuation as well as $CO_{2(\varepsilon_p-alk)}$, and allowed for uncertainty estimates to be site-ambivalent.

The result is shown in Figure 6c,d, and suggests that the 7 $\mu molL^{-1}$ break point remains valid. The absolute value of $r^2$ is reduced, even at higher $[CO_2]_{(aq)-predicted}$, but this would be expected given the addition of uncertainty in age model, as the published age is most likely to align with the ice core. Given the rapid rate of change at deglaciations, this effect is likely to be particularly pronounced in this dataset as many records have high temporal resolution around deglaciations in order to attempt

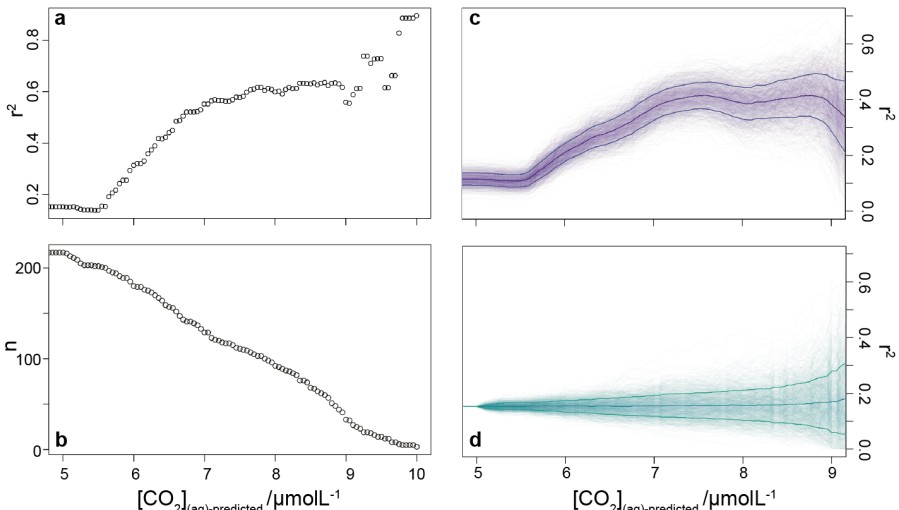

**Figure 6.** Coefficient of determination (panel a) of a reducing sample of all compiled $CO_{2(\varepsilon_p-alk)}$ (Table 1) vs the time-equivalent estimate from ice core records (Bereiter et al. (2015); Table 2). The sample reduces stepwise by 0.05 $\mu molL^{-1}$, and the number of records in each subsample is shown in panel b. Panel c shows a 1000 member Monte Carlo analysis, whereby uncertainty in $CO_{2(\varepsilon_p-alk)}$ and age is considered, as detailed in the text. Panel d shows a similar 1000 member Monte Carlo analysis, but with random sampling of the whole $CO_{2(\varepsilon_p-alk)}$ population so that the number of samples is equivalent to the dataset shown in panel c, ie the size of the sample follows that shown panel b. Means and one $\sigma$ uncertainties are shown as the bold lines.

to resolve them. Any small age model offset introduced by the error modelling in these intervals also clearly has the potential to induce large differences between the $CO_{2(\varepsilon_p-alk)}$ and ice core values. Figure 6c,d clearly demonstrates that it is the filtering by $[CO_2]_{(aq)-predicted}$ rather than any spurious correlations which determine the shape of the data in Figure 6a.

## 4   Discussion

The plateau in $r^2$ in Figures 6a and 6c suggest that below a $[CO_2]_{(aq)-predicted}$ of $\sim 7$ $\mu molL^{-1}$ $CO_{2(\varepsilon_p-alk)}$ is no longer as good a predictor of ice core $CO_2$ as when $[CO_2]_{(aq)-predicted} > 7$ $\mu molL^{-1}$. This is clear from comparing the relationship between samples where $[CO_2]_{(aq)-predicted} < 7$ $\mu molL^{-1}$ with those where $[CO_2]_{(aq)-predicted} > 7$ $\mu molL^{-1}$ in Figure 8. Here the $r^2$ for the former of 0.15 is substantially less than the latter of 0.55. I suggest that this is because below this threshold, the fundamental assumption of $CO_{2(\varepsilon_p-alk)}$; that carbon is passively taken up by haptophytes, no longer holds true. One obvious explanation for why this would be the case is that at low levels of $[CO_2]_{(aq)}$ haptophytes have to rely more on active up take of carbon via CCMs in order to satisfy metabolic demand. Similar behaviour has been recognised in some culture studies (Laws et al., 1997, 2002; Cassar et al., 2006), with some evidence that the diatom *Phaeodactylum tricornutum* has a similar CCM threshold of 7 $\mu molL^{-1}$ (Laws et al., 1997). Whilst the evidence for the mechanism of CCM is poorer for coccolithophores than

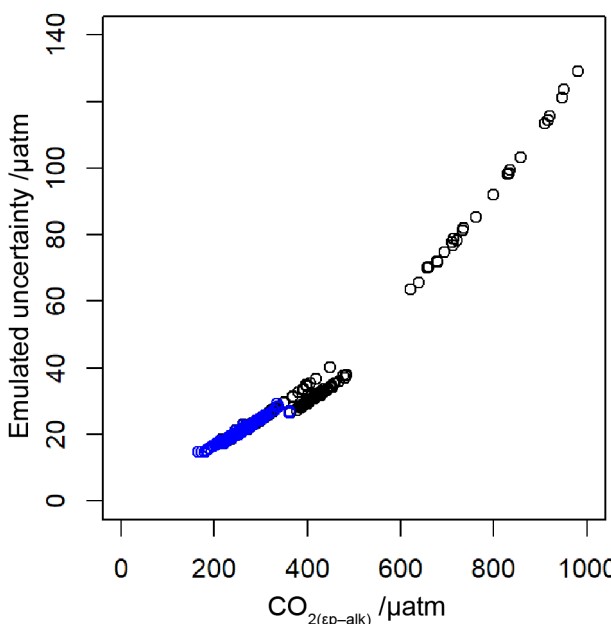

**Figure 7.** Emulated uncertainty in $CO_{2(\varepsilon_p-alk)}$, generated by running Monte Carlo uncertainty models for all sites in Table 1 applying the same approach to uncertainty as Badger et al. (2013a, b). Estimates used in this study are highlighted in blue.

it is for diatoms, any CCM would be expected to compromise the $CO_{2(\varepsilon_p-alk)}$ proxy, either by increased supply of $[CO_2]_{(aq)}$, further carbon isotopic fractionation effects during carbon transport or both (Stoll et al., 2019).

By applying a threshold value for $[CO_2]_{(aq)-predicted}$ of 7 µmolL$^{-1}$ to the published records (Table 1) values of $CO_{2(\varepsilon_p-alk)}$ which are influenced by active CCMs can be eliminated. Recognition of this new threshold value of $[CO_2]_{(aq)-predicted}$ allows for a new record of Pleistocene $CO_{2(\varepsilon_p-alk)}$ to be compiled. This compilation then much better replicates the glacial-

interglacial pattern of $CO_2$ change over the last 260 Ka (Figure 9). Whilst this present compilation does rely on ice core $CO_2$ records to estimate $[CO_2]_{(aq)-predicted}$, and therefore has little direct utility as a $CO_2$ record, it does demonstrate that recognition of a threshold response allows accurate $CO_2$ reconstruction using $CO_{2(\varepsilon_p-alk)}$. This may represent the point at which isotopic effects of CCMs (plausibly through increased CA activity or $HCO_3^-$ dehydration to meet C demand) overwhelms the assumptions of the $CO_{2(\varepsilon_p-alk)}$ proxy. This, and the behaviour shown in Figures 6a and 6c suggests that from the standpoint

of the $CO_{2(\varepsilon_p-alk)}$ proxy CCMs may effectively be considered either active or not, and that when $[CO_2]_{(aq)}$ is plentiful passive uptake dominates, at least sufficiently in most oceanographic settings that $CO_{2(\varepsilon_p-alk)}$ can accurately record atmospheric $CO_2$ concentration. This implies that if areas of the ocean (or intervals of time) with low $[CO_2]_{(aq)}$ can be avoided, accurate reconstructions of atmospheric $CO_2$ concentration can be acquired using $CO_{2(\varepsilon_p-alk)}$.

As $[CO_2]_{(aq)}$ is effected by both SST via the temperature dependance of the Henry's law constant and atmospheric $CO_2$ con-

centration, for $CO_{2(\varepsilon_p-alk)}$ to be effective in reconstructing atmospheric $CO_2$ concentration, areas of warm water (i.e. tropical

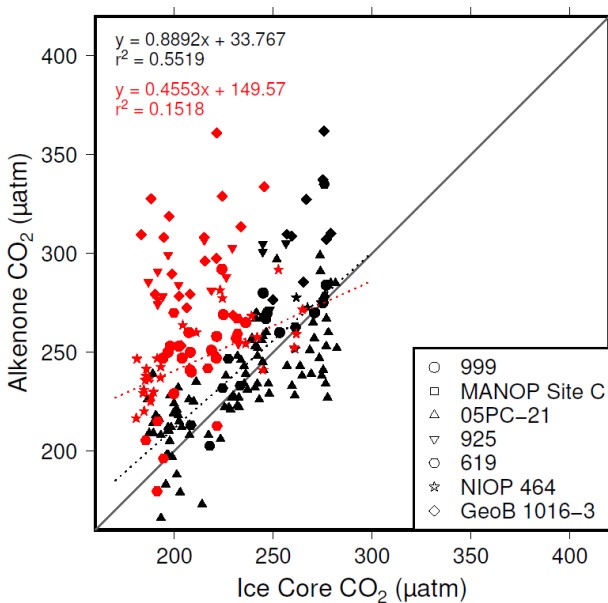

**Figure 8.** Correlations between $CO_{2(\varepsilon_p-alk)}$ and ice core $CO_2$ where $[CO_2]_{(aq)-predicted} > 7$ µmol$L^{-1}$ (black symbols) and $[CO_2]_{(aq)-predicted} < 7$ µmol$L^{-1}$ (red symbols).

or shallow shelf regions) under relatively low atmospheric $CO_2$ concentration must be avoided. However, as the atmospheric $CO_2$ control renders the global surface ocean sufficiently replete in $[CO_2]_{(aq)}$ at Pliocene-like levels of atmospheric $CO_2$ concentration and above (Martínez-Botí et al., 2015) at all but the warmest surface ocean temperatures, $CO_{2(\varepsilon_p-alk)}$ is likely to be a reliable system for most of the Cenozoic. It is only in the Pleistocene that atmospheric $CO_2$ concentration is low enough

for CCMs to be widely active accross the surface ocean, with the low $CO_2$ glacials providing the most difficulty (Badger et al., 2019). This finding aligns well with evidence that CCMs developed in coccolithophores as a reponse to declining atmospheric $CO_2$ concentration through the Cenozoic, and were developing in $[CO_2]_{(aq)}$–limited parts of the ocean in the late Miocene at the earliest, and likely not widespread until the Plio-Pleistocene (Bolton et al., 2012; Bolton and Stoll, 2013).

There have been recent attempts to correct for CCMs in $CO_{2(\varepsilon_p-alk)}$-based reconstructions of atmospheric $CO_2$ concen-

tration s (Zhang et al., 2019; Stoll et al., 2019; Zhang et al., 2020). However, these assume that CCMs are always active, and crucially do not fundamentally break the relationship between $\varepsilon_p$ values and atmospheric $CO_2$ concentration. However if this is not the case, and the relationship between $\varepsilon_p$ values and atmospheric $CO_2$ concentration fails at Pleistocene levels of atmospheric $CO_2$ then Pleistocene records cannot be used to develop corrections of $CO_{2(\varepsilon_p-alk)}$ to be applied throughout the Cenozoic. If, as suggested by the analyses presented here, CCMs *only* act at low $[CO_2]_{(aq)}$, and largely only in conditions

prevalent throught the late Pliocene and Pleistocene, it is plausible that corrections based on Pleistocene records could over-compensate for CCMs in the rest of the Cenozoic, when the assumption of passive carbon uptake inherent in $CO_{2(\varepsilon_p-alk)}$ as traditionally applied may still be valid.

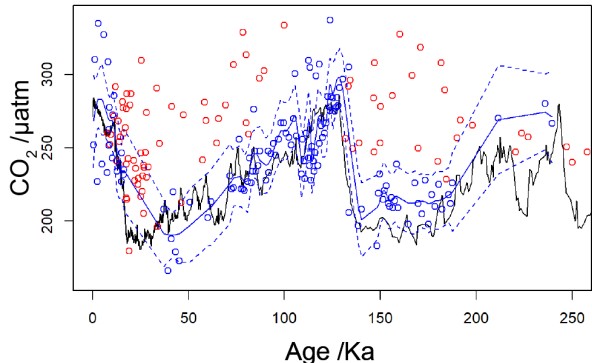

**Figure 9.** Revised compilation of Pleistocene $CO_{2(\varepsilon_p-\mathrm{alk})}$ vs ice core records. The compiled published records (Table 1) are shown as circles, coloured red where $[CO_2]_{(\mathrm{aq})-\mathrm{predicted}}$ is below a threshold of 7 $\mu mol L^{-1}$, and blue where $[CO_2]_{(\mathrm{aq})-\mathrm{predicted}} > 7\ \mu mol L^{-1}$. The solid blue line is a loess filter (span 0.1) through the $[CO_2]_{(\mathrm{aq})-\mathrm{predicted}} > 7\ \mu mol L^{-1}$ values, with 95 % confidence intervals (dashed blue line). The black line is the ice core compilation of Bereiter et al. (2015) (Table 2).

## 5 Conclusions

Reconstructions of past atmospheric $CO_2$ concentration with proxy tools like $CO_{2(\varepsilon_p-\mathrm{alk})}$ are critical for understanding how the Earth's climate system operates, as long as the tools used can be relied upon to be accurate and precise. This re-analysis of existing Pleistocene $CO_{2(\varepsilon_p-\mathrm{alk})}$ records reveals that below a critical threshold of $[CO_2]_{(\mathrm{aq})}$ of 7 $\mu mol L^{-1}$ the relationship between $\delta^{13}C_{\mathrm{alkenone}}$ and atmospheric $CO_2$ concentration breaks down, plausibly because below this threshold haptophytes are able to actively take up carbon using CCMs in order to satisfy metabolic demand.

Although reconstructing the low levels of atmospheric $CO_2$ concentration in the Pleistocene glacials and areas of the global ocean where $[CO_2]_{(\mathrm{aq})}$ is less than 7 $\mu mol L^{-1}$ will be impossible, for much of the Cenozoic, the $CO_{2(\varepsilon_p-\mathrm{alk})}$ proxy retains utility. If care is taken to avoid regions and oceanographic settings where $[CO_2]_{(\mathrm{aq})}$ is expected to be abnormally low, $CO_{2(\varepsilon_p-\mathrm{alk})}$ remains an important and useful proxy to understand the Earth system.

*Code and data availability.* This paper relies exclusively on previously published data, available with the original papers and in publicly available repositories. An R notebook supplement is available alongside this manuscript, along with datafiles which allow full replication of all analyses performed.

*Author contributions.* MPSB conceived the study, designed the methodology, analysed the data, prepared the figures and wrote the manuscript (conceptualization, formal analysis, investigation, methodology, vizualization, writing - original draft, review and editing)

*Competing interests.* MPSB declares that he has no conflict of interest

*Acknowledgements.* I am grateful to Gavin Foster and Tom Chalk for frequent and stimulating discussions on alkenone paleobarometry. I
thank all authors who made full datasets available online. I thank Kirsty Edgar for comments on various drafts, and the two anonymous
reviewers whose comments greatly improved this manuscript.

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
