# Peer review of "Alkenone isotopes show evidence of active carbon concentrating mechanisms in coccolithophores as aqueous carbon dioxide concentrations fall below 7 $\mu$ molL-1"

_Biogeosciences, 2020_

## Short Comment (SC1) · 15 Oct 2020

This manuscript is well-written and I expect that it will be a valuable addition to the on-going discussion on CCMs in phytoplankton and coccolithophores in particular. There is something that I do not understand though. Based on Figure 2 I deduce that all of the sites used - with the exception of this Manop C site - have the same present-day CO2 partial pressure anomaly. In addition, all of the sites are within 0-30 N/S. This then makes me conclude that the calculation of [CO2]aq using the ice-core pCO2 estimates should primarily depend on SST and salinity estimates back in time right? I personally

cannot see how SST and salinity back in time would be so much different for these very similar sites. I find this difficult to believe and it would be helpful if you could show the maps of the SST and salinity values used to calculate [CO2]aq. Also, I find it a bit of an easy statement to make that low CO2 should trigger the CCMs in coccos if recent papers have convincingly shown that light level and nutrient availability are main triggers of CCMs (e.g. Wilkes et al., 2019). Last, I find it a bit of an overstatement to say that as CO2 equilibrium should be better maintained at high atmospheric CO2 into the Cenozoic, and as CCMs should likely not operate at high pCO2; we can thus use the alkenone-pCO2 proxy confidently throughout the whole Cenozoic. Could you discuss the uncertainty in this extrapolation? After all, pCO2 through glacial-interglacial times never exceeds 280ppm. Furthermore, what about coccolith evolution through time? I hope these comments are helpful.

---

## Author Comment (AC1) · 19 Oct 2020

Many thanks for your kind comments on the manuscript. I hope if nothing else this manuscript spurs further discussion and research into this matter.

You are right that there are not large differences between equilibrium at the different sites, although there are some differences, which are lost in the scale needed for a global figure like figure 2. The more important site-to-site changes here are, as you deduce the different SSTs and SST history of each site, which, through Henry's law, have

a substantial impact on [CO2](aq) absolutely and through time. For SST over time I use the published records used for each of the records I have analysed. Salinity is much more difficult to estimate through time, and records of salinity was not easily to hand for each site so I kept S constant at each site. The combination of SST and disequilibrium differences between the site mean that through the interval that the records cover predicted [CO2](aq) varies across the sites from 5 to 10.6 umolL-1, as shown in Figure 4. The SST records are available for further interrogation in the original publications, and are compiled (alongside predicted [CO2](aq)) in the supplement.

I too am intrigued as to how my findings fit into the Wilkes et al., (2019) framework. The bloom-forming nature of the coccolithophores in particular do make it difficult to translate batch and chemostat cultures varying light and nutrient levels from the lab to the oceans, as both will vary through a bloom event. The Wilkes et al., (2019) model also doesn't rule out that kinetic, [CO2](aq) dependence of ep still occurs, and do note several examples in the ocean where that does appear to be the case. It may be that nutrients, light and CO2 availability all play a part, and my study picks out the times where CO2 availability becomes the most important. If ep is fully controlled by nutrients and light levels alone, then the good fit of the alkenone data to the ice core in Figure 10 needs explanation, as does the observation of Bolton et al., (2012) and Bolton and Stoll (2013) that CCMs only become widespread once we reach the lower atmospheric CO2 worlds of the Plio-Plesistocene compared to the rest of the Cenozoic, perhaps as an evolutionary response.

My suggestion that alkenone-pCO2 can be used confidently through the Cenozoic is only on the basis that the very low [CO2](aq) conditions observed in the Pleistocene ocean (particularly) in the glacials become less likely in the higher-CO2 world of the Neogene. I cannot (and do not) rule out that, if CCMs do become active at low [CO2](aq), the could be occasions in the Cenozoic where these conditions are seen at certain sites, but if the atmosphere is overall more replete in CO2, this should be less likely. For example, if you do a quick estimation of global ocean [CO2](aq) in a

warmer world (+ 3 degC) with atmospheric CO2 at Pliocene-like levels is no region of the ocean where [CO2](aq) is expected to be below 7 umolL-1 (Figure 1).

Figure 1: Estimated global [CO2](aq) for a Pliocene-like world with atmospheric CO2 close to 400 ppm and SST 3 degrees warmer than today. (Based on data from Taka-hashi et al., (2014)).

[Figure]

**Fig. 1.**

---

## Short Comment (SC3) · 20 Oct 2020

Many thanks for the kind and swift response to my comments, and the answers to my questions.

I am also intrigued by the results as displayed in Figure 10 of the manuscript. Before I continue, I want to emphasize that I am no expert on the coccolithophore ep-based pCO2 proxy. I hope that I can first and foremost be a helpful objective new participant in this very important field of climate research.

Regarding the Wilkes et al. (2019) framework; I agree that nutrients, lights and CO2 availability all play a part, and that it may indeed be accurate that CO2 availability plays a dominant part in times when CO2 availability drops to the lowest concentrations known in Earth's history. That being said, I do want to exert caution. There is a merit in the statistical approach undertaken in this study, but please do not forget the other pieces of information in the separate data-sets. The relationship between the b-factor and phosphate concentrations is - after all - problematic, as stated in the study (e.g. Zhang et al., 2020). In short, this relationship may not hold for every phytoplankton community, at every location on the planet. For example: yes, phosphate concentrations are greater in the Peru upwelling region, but so is algal growth, and thus light availability at this location may exert a significant control on ep, perhaps more so than [CO2]aq. In this context, I think the study can benefit from a discussion of the Wilkes et al. (2019) framework, if only to get all its readers on the same page.

After reading your reply, I also think that the exclusion of the Manop C site could use some more discussion. I find it interesting that the entire data-set seems to be offset by a constant number. Is this truly due to a change in CO2 equilibrium? Or are other factors playing a role here? Are oceanographic frontal changes to blame? This brings me to what I wrote before. When I look at the data-set from Site 619, it seems to agree quite well with ice core pCO2 reconstructions, even though it seems to have witnessed times with [CO2]aq below 7 umol/L. I know that the whole point of your study is the statistical approach, but a little more discussion on the sites itself; perhaps in combination with the Wilkes et al. (2019) framework, could be useful.

To summarize, yes, the good fit in Figure 10 could indeed be related to the activation of CCM below 7 umol/L, but it could also not be. Some more discussion of why it could not be would be good I think.

---

## Referee Comment (RC1) · Anonymous Referee #1 · 24 Oct 2020

—Summary and Recommendation —

The manuscript entitled "Alkenone isotopes show evidence of active carbon concentrating mechanisms in coccolithophores as aqueous carbon dioxide concentrations fall below 7 $\mu$molL$-1$" explores the lower limitations of the alkenone-based pCO2 proxy. The proxy relies on the assumption that the isotopic composition of alkenone compounds reflect the abundance of aqueous CO2 in its growth water which is passively diffused in the cell. However, there has always been an underlying concern of this assumption, as alkenone-producing algae have been shown to have carbon concen-

trating mechanisms, a concern given that there are different mechanisms involved and the significant difference in the isotopic composition of bicarbonate versus aqueous $CO_2$. To explore the potential influence of carbon concentrating mechanisms on the alkenone-based proxy, the author compiled 6-7 studies using the exact same calculation methods, and then compared these indirect proxy-based estimations with direct ice core measurements from the same time period (with consideration to age uncertainties). The author found that the relationship between the alkenone-based proxy and the ice core data breaks down with decreasing aqueous $CO_2$, with an $r^2$ dropping below 0.5 when aqueous $CO_2$ is below 0.7 $\mu$molL$-1$.

This manuscript represents a substantial contribution to scientific progress in the field of biogeochemistry, as well as paleoclimate, by addressing the physiological limitations of this widely used $CO_2$ proxy. Overall, the scientific quality is high: the methods are robust, the logic is very clear, and there is consideration to the body of literature on this topic. Finally, the presentation quality is also excellent. The manuscript is clear, concise, and well-structured; in fact, this is one of the most easy-to-follow explanations of the alkenone-based proxy that I have read. Furthermore, the figures are well-made and well-suited to the text. I highly recommend this manuscript for publication with minor revisions. Below are some questions and criticisms to be addressed before this manuscript is accepted.

—General Comments—

The fact that these many studies have now all been calculated using the exact same methods should be included in the Abstract and Introduction (re: Line 100-101), as this alone is important for the community. Consider moving (or repeating) Line 173-175 in the Abstract and/or Introduction, as this is also an important point of this study.

The Introduction needs to establish some key topics. Most importantly, there is almost no introduction of carbon concentrating mechanisms, which is the entire crux of the study. In particular, it is important to establish how they work, the different kinds of

mechanisms (e.g. external or intracellular carbonic anhydrase, $HCO_3-$ transport), and why CCMs are concerning for the proxy (e.g. isotopic difference between bicarbonate and $CO_2$, different fractionation pathways). It might also be useful to further discuss alkenone-producers, including which have CCMs and possible differences among alkenone-producers. In the studies compiled here, is there an idea of which producers are present (i.e. coccoliths preserved in the sediment)? Any size-corrections?

Throughout the manuscript, there is almost no mention of the other possible causes for the breakdown relationship between epsilon p and dissolved $CO_2$. Although CCMs are certainly a strong possibility, the author should also consider changes to the "b" factor or environmental conditions. These need to be addressed in the Introduction and later on in the Discussion. Has the author considered looking for indicators of e.g. upwelling (which would increase the availability of aqueous $CO_2$) or the BIT index (indicating increased terrestrial input, effecting nutrient availability or species composition change)? If possible, include site information such as estimated depth and distance from coast, as this is important to interpreting the results.

Following up on this point, Lines 187-188, the author states that areas of warm water (i.e. tropical or shallow shelf regions) should be avoided. However, essentially all of the sites used in this study are tropical low latitude (30° N to 30° S) and all look quite close to the continents, likely to be shallow. Is there a possibility that the sites are the issue (e.g. warm waters, upwelling, growth factors), not necessarily the proxy mechanisms? This Lines 187-188 statement also seems at odds with Lines 50-51, where the author states that low latitude, gyre sites are likely more oceanographically stable. In the Methods, the author discusses the use of phosphate to determine b. Could the author briefly include why they are not considering the findings of Zhang et al. (2019; 2020)? Any possibility of comparing with $\delta$15N values? (see Andersen et al. 1999 in Use of Proxies in Paleoceanography, Ch. 19, 469–488). The author uses an uncertainty of 11% on b, though the actual possible range is extremely large, ranging from about 50 to 200, as seen in Pagani (2014).

[Figure]

In the Results, Line 102-107, the author suggests that CCMs only come actively start pumping under a certain low-CO2 threshold. This needs further support/references, given that the literature has shown that CCMs are quite complex in their function and varied among species (e.g. Reinfelder, 2011, Annual Review of Marine Science). Line 105-107: rephrase as it is currently misleading. If we assume that carbon concentrating mechanisms are prevalent, then the proxy would perform least well under low CO2 concentrations. However, as currently phrased, if the alkenone-based proxy "relies on the assumption of a purely diffusive uptake of carbon", then actually, we would expect the proxy to perform the same as it does at any level.

—Figures and Tables—

Overall, these are very impressive figures. There is a lot of repetition throughout (e.g. Fig. 3 and Fig. 10 are nearly identical) but is effective for telling this story.

Table 1: I suggest adding a column with the approximate age ranges for each site. I would also suggest giving this table some kind of structured order (maybe by latitude?)

Fig. 2: NIOP 464 should be a star but is expressed as a square (there are two square symbols). It is difficult to distinguish these shapes, as they are very small (e.g. the circle and hexagon look identical unless I zoom in). Please add the site numbers next to the locations or at the very least, make the symbols more distinct with an accompanying legend on the figure.

Fig. 3: Consider including the site symbols.

Fig. 6: Consider indicating the breakdown point of proxy vs ice core (e.g. using a dashed line)

Fig. 7: Between ∼400-500 [CO2(ep-alk)/uatm], why are there two separate estimates for [emulated uncertainty/uatm]?

Fig. 8: Because Fig 6 and 8 are so similar and are constantly discussed together, it might be worth combining these into one figure with 4-quadrants (instead of two figures

with 2-quadrants).

Fig. 10: Gorgeous!

—Minor comments—

There are numerous spelling and grammatical issues throughout the manuscript, for example: Line 20 (isotopic), Line 26 (Noelaerhabdaceae), Line 106 (diffusive), Line 145 (calculated), Fig. 4 (estimate). Some of the references also have incorrect the incorrect doi or link. Please revise.

Line 20 (and throughout): Add "stable" before carbon isotopic composition

Line 28 (and throughout): Add "concentrations" when regarding atmospheric CO2

Line 29: Consider including other CO2 proxies, e.g. paleosols, leaf gas exchange models, liverworts, C3 plants

Line 33: I would not consider less than 400 uatm "moderate to low". The author may avoid the subjective term altogether by rephrasing to ". . . at atmospheric CO2 concentrations below 400 uatm of the Pleistocene".

Line 36: Although the Super et al. (2018) SST reconstruction is incredibly useful, I would not consider the Miocene a "resolved" issue. The Miocene CO2 is a highly debated topic at the moment.

Line 46 (and throughout): Add "values" after all $\delta$13C

Line 118-121: Remove "such as" throughout. Here, the author includes every single site, so there's no need to express them as examples.

Line 143-150: Please break down into several sentences, quite difficult to read.

Line 167: Repetitive sentence

Line 175: Remove "even"

Line 196: "Recent" what?

---

## Referee Comment (RC2) · Anonymous Referee #1 · 24 Oct 2020

When opening the supplemental files, the PDFs open just fine but unfortunately, the data spreadsheets do not. Please provide these files to be reviewed. Thank you.

---

## Referee Comment (RC3) · Anonymous Referee #2 · 30 Oct 2020

The manuscript by Badger presents the capability of the haptophyte alkenone proxy to reconstruct past atmospheric CO2 concentrations compared to the ice core CO2 records during the Pleistocene, based on alkenone CO2 reconstructions of numerous sites. Deviations that the proxy exhibit from the ice core records are attributed to the activation of carbon concentrating mechanisms (CCMs) in the haptophytes during C acquisition under low CO2 environments, and the author therefore concludes this proxy may not be valid during periods of expected low atmospheric CO2 conditions (corresponding to below ~7 umol/L of aqueous CO2) . Overall, this work is very relevant

to the field and well written, especially the inclusion of multiple sites, the comparison to the ice core record, and the overall very clear presentation. However, the statement that deviations between the proxy and ice core record is due to the activation of CCMs is a bit too short and does not consider all the knowledge we have on CCMs in haptophytes. This can be improved by sufficient introduction into the subject and more discussion on possible explanations on why this deviation between the ice core record and the alkenone proxy may exist. After addressing the following points (minor revisions) the manuscript is worthy of publication.

General comments

As mentioned above, the main issue I have is the statement that the activation of CCMs is the sole explanation between the offset in CO2 reconstructions between the alkenone proxy and ice core record. There is no evidence that since the development of CCMs in haptophytes (which may have occurred during the late Miocene – early Pliocene) these CCMs could be turned on and off. On the contrary, a study from Van de Waal et al. 2019 (L&O Letters) suggests that haptophyte CCMs (measured in present day haptophytes) are not so adjustable even in high CO2 environments. Although this is from present day haptophytes, no mention of such findings is made, even though the data presented here is closer in age to the present-day haptophytes than those from late Miocene.

The main conclusion of the paper also revolves around CCMs, but this topic is hardly introduced or properly explained in the introduction and explored in the discussion. This can certainly be improved. CCMs also comprise various mechanisms of acquiring C and it can be explored how alterations in these strategies may compromise alkenones being a reliable proxy for atmospheric CO2. It may have something to do with increased uptake of HCO3 relative to CO2, but no mention of this is made. Just stating CCMs are turned on or off is a bit oversimplified, especially since there is not a lot of evidence for this.

A bit more emphasis on the comparison of the alkenone proxy to the ice core record may be made in the title and in the abstract, as it is very nice that data from multiple sites are combined and also demonstrates the pitfalls of this proxy.

Specific comments

There are still a few mistakes and a few awkward sentences in the text.

Line 17-21 Quite a long and confusing sentence with conjugations that do not fit.

Line 30 two times “to” close together, reads a bit odd

Line 32 However, recent. . .

Line 35 Differences

Line 36 appear

Line 38 operation instead of action

Line 40 it is written here as though CCMs are usually not active and sometimes get active, but it is usually the case that they are active from what we know of present-day phytoplankton

Line 58 take up instead of uptake

Line 59 were instead of I

Line 61, but also Line 84 here you state that additional corrections from the original records were removed, but you accounted for that in the fractionation with the “b” term, right? How exactly is this term calculated for all the sites?

Line 106 worse instead of less well

Line 108 haptophytes

Line 143 twice that

Line 153 were instead of was

Line 172 take up instead of uptake

Line 175 what do you mean here? The study you did or the one from Laws? Not clear from sentence structure, although I assume you mean your study as you refer to alkenones. If so, I would not state it like this, as you only look at sedimentary records which is not clear behavior of activation of CCMs.

Line 178 influenced

Line 182 not sure if this is necessarily a CCM threshold or a switch maybe from one of mechanisms of the CCM (for instance a switch from CO2 to HCO3uptake)

Line 187 maybe also state how SST influences aqueous CO2, as this is not yet mentioned.

Line 196 word missing after "recent" Line 203 critical for

Line 204 as long as

Line 206 take up

Line 209 comma after Cenozoic

Line 210 may be abnormally, or is expected to be abnormally

---

## Author Comment (AC2) · 3 Dec 2020

Thankyou to the reviewer for their kind words about the manuscript and the contribution it represents. I am happy to include the suggestions the reviewer makes in a revised manuscript. A common theme in both reviews received is the need for greater and more detailed introduction of carbon concentrating mechanisms in haptophytes, I am happy to include a new section in the introduction covering this, and this will allow further discussion to be added later in the manuscript as well. Alkenone producers in the modern ocean are relatively well known, and I will include a discussion of this, as

well as a discussion of what is known about past alkenone producers.

In this manuscript I left much of the discussion of the possible causes for the breakdown in the relationship between epsilon p and dissolved CO2 short, as although this work does suggest where and when in the ocean this breaks down, and it makes sense that CCMs would be at play, it is perhaps beyond this work to confirm that. I can however expand this discussion. The strength, I believe, of this work is the power of combining the multiple records and treating them the same. Unfortunately, not all published records have suitable indictors of upwelling and runoff (like BIT), so this level of analysis may be beyond this work and a fruitful avenue of further information. In a revised manuscript I can include what information is available, especially with records to depth and distance from the coast.

The sites are in relatively diverse settings, with ODP sites 925 and 999 considered open ocean sites, and the analysis I show suggests that it isn't one site or another which performs better, rather the [CO2](aq), as can be seen from Figure 4. The diversity of sites will be clearer once I include water depth, which I can add to Table 1.

Similar to the lack of BIT for all Sites, few have nitrogen isotopes available, which precludes a full assessment. The findings of Zhang et al (2019, 2020) are an interesting avenue for further developing the proxy, but I would caution that some of those analyses start from the premise that you can use Pleistocene records to re-calibrate "b", and my analysis would suggest that for many of these records there are times when alkenone isotopes are no longer sensitive to atmospheric CO2 changes, which may require a reassessment of some of that work.

CCMs are indeed quite complex, and I am happy to add futher nuance to the discussion of them, my point with "relies on the assumption of a purely diffusive uptake of carbon" is that the proxy as currently applied is far too simplistic in the low [CO2](aq) situations where CCMs may dominant, assuming as it does that CCMs are not important. I can revise this statement.

There was a problem with the zip file which contained the supplemental data which as been resolved.

---

## Author Comment (AC3) · 3 Dec 2020

Many thanks to the reviewer for their kind word on the manuscript and constructive comments and suggestions. I am happy to make all the minor revisions suggested in a revised manuscript. In common with the other reviewer, this reviewer suggests a need for enhanced introductory and discussion text about CCMs and I am happy to make these additions in a revised manuscript.

The results of Van de Waal et al (2019) are certainly intriguing, but it is difficult to be

sure how these results apply to much of the interval of my study, as the most relevant haptophyte studied, Emiliania huxleyii, is a relatively recent species and has evolved (and in particular become dominant) within the relatively low CO2 conditions of the Pleistocene.

---

## Author Response (AR1)

**Author response for "Alkenone isotopes show evidence of active carbon concentrating mechanisms in coccolithophores as aqueous carbon dioxide concentrations fall below 7 µmol L⁻¹" by Marcus P. S. Badger**

Anonymous Referee #1

Many thanks to the reviewer for the kind words on the manuscript and the constructive comments and suggestions.

> The fact that these many studies have now all been calculated using the exact same methods should be included in the Abstract and Introduction (re: Line 100-101), as this alone is important for the community. Consider moving (or repeating) Line 173-175 in the Abstract and/or Introduction, as this is also an important point of this study.

I have included a line to this effect in the Abstract (lines 7-8; line numbers throughout refer to the latexdiff file for easy reference)

> The Introduction needs to establish some key topics. Most importantly, there is almost no introduction of carbon concentrating mechanisms, which is the entire crux of the study. In particular, it is important to establish how they work, the different kinds of mechanisms (e.g. external or intracellular carbonic anhydrase, HCO3⁻ transport), and why CCMs are concerning for the proxy (e.g. isotopic difference between bicarbonate and CO2, different fractionation pathways). It might also be useful to further discuss alkenone-producers, including which have CCMs and possible differences among alkenone-producers. In the studies compiled here, is there an idea of which producers are present (i.e. coccoliths preserved in the sediment)? Any size-corrections?

I now include a new section in the Introduction outlining what is understood about CCMs in coccolithophores (new section 1.1, lines 45 -79) and further lines on alkenone producers in the introduction (lines 30-33). As noted within the methods I use a common methodology for all records which does strip away some secondary corrections (lines 99-104), for size corrections in particular, there is not a consensus about how to perform these (see discussion in Badger et al 2019), and very few records had lith size records available.

> Throughout the manuscript, there is almost no mention of the other possible causes for the breakdown relationship between epsilon p and dissolved CO2. Although CCMs are certainly a strong possibility, the author should also consider changes to the "b" factor or environmental conditions. These need to be addressed in the Introduction and later on in the Discussion. Has the author considered looking for indicators of e.g. upwelling (which would increase the availability of aqueous CO2) or the BIT index (indicating increased terrestrial input, effecting nutrient availability or species composition change)?

As noted in the discussion phase, and similar to lith size, not all records have suitable BIT or upwelling records to compare to, and the strength of the analysis presented is the ability to treat all records equally. I have expanded my discussion of 'b' corrections in the discussion (lines 260-265) however as noted there the recent attempts to correct the 'b' term has relied on the assumption that the proxy system is working in the Pleistocene, but with reduced sensitivity or secondary corrections. My work here suggest that for some of the Pleistocene at some sites the proxy system breaks down, and so this sort of correction is inappropriate. I have clarified this point (line 262-3).

> If possible, include site information such as estimated depth and distance from coast, as this is important to interpreting the results.
> Following up on this point, Lines 187-188, the author states that areas of warm water (i.e. tropical or shallow shelf regions) should be avoided. However, essentially all of the sites used in this study are tropical low latitude (30_N to 30_S) and all look quite close

to the continents, likely to be shallow. Is there a possibility that the sites are the issue (e.g. warm waters, upwelling, growth factors), not necessarily the proxy mechanisms? This Lines 187-188 statement also seems at odds with Lines 50-51, where the author states that low latitude, gyre sites are likely more oceanographically stable.

This information now included in Table 1, from which it can be seen that although many of the sites are low latitude (this is noted lines 87-9) they are otherwise quite diverse and range in water depth and distance from the coast.

In the Methods, the author discusses the use of phosphate to determine b. Could the author briefly include why they are not considering the findings of Zhang et al. (2019; 2020)? Any possibility of comparing with _15N values? (see Andersen et al. 1999 in Use of Proxies in Paleoceanography, Ch. 19, 469–488).

15N data is not available for most sites but could be of interest in future work, the reasons for not considering Zhang (2019; 2020) is, as noted above, because this work potentially makes the analysis of Zhang et al (2019;2020) incorrect, as noted in new lines 260-265.

In the Results, Line 102-107, the author suggests that CCMs only come actively start pumping under a certain low-CO2 threshold. This needs further support/references, given that the literature has shown that CCMs are quite complex in their function and varied among species (e.g. Reinfelder, 2011, Annual Review of Marine Science). Line 105-107: rephrase as it is currently misleading. If we assume that carbon concentrating mechanisms are prevalent, then the proxy would perform least well under low CO2 concentrations. However, as currently phrased, if the alkenone-based proxy "relies on the assumption of a purely diffusive uptake of carbon", then actually, we would expect the proxy to perform the same as it does at any level.

I now include a much fuller discussion of CCMs in the Introduction (new section 1.1. lines 45-79) and have expanded the discussion section on this point (new and revised material lines 229-249) and have revised the phrase in question (line 243)

Table 1: I suggest adding a column with the approximate age ranges for each site. I would also suggest giving this table some kind of structured order (maybe by latitude?)

Done (new table 1)

Fig. 2: NIOP 464 should be a star but is expressed as a square (there are two square symbols). It is difficult to distinguish these shapes, as they are very small (e.g. the circle and hexagon look identical unless I zoom in). Please add the site numbers next to the locations or at the very least, make the symbols more distinct with an accompanying legend on the figure.

Fixed and site labels added (new Figure 2)

Fig. 3: Consider including the site symbols.

Done. (new Figure 3)

Fig. 6: Consider indicating the breakdown point of proxy vs ice core (e.g. using a dashed line)

I try to avoid adding guides to figures – if the eye needs guiding often the point is not as strong as suggested, I hope that this point stands out sufficiently.

Fig. 7: Between _400-500 [CO2(ep-alk)/uatm], why are there two separate estimates for [emulated uncertainty/uatm]?

This is just the result of the interaction of different parameters in the different datasets (I think the lower one is at a lower temperature).

> Fig. 8: Because Fig 6 and 8 are so similar and are constantly discussed together, it might be worth combining these into one figure with 4-quadrants (instead of two figures with 2-quadrants).

This is a good idea, which I have implemented in a new Figure 6.

> There are numerous spelling and grammatical issues throughout the manuscript, for example: Line 20 (isotopic), Line 26 (Noelaerhabdaceae), Line 106 (diffusive), Line 145 (calculated), Fig. 4 (estimate). Some of the references also have incorrect the incorrect doi or link. Please revise.

Done (see marked-up file).

> Line 20 (and throughout): Add "stable" before carbon isotopic composition
> Line 28 (and throughout): Add "concentrations" when regarding atmospheric CO2

Done (see additions marked throughout the marked-up file)

> Line 29: Consider including other CO2 proxies, e.g. paleosols, leaf gas exchange

As the two marine proxies are most comparable in terms of temporal usage (both range and resolution) I have kept at these two.

> Line 33: I would not consider less than 400 uatm "moderate to low". The author may avoid the subjective term altogether by rephrasing to ": : : at atmospheric CO2 concentrations below 400 uatm of the Pleistocene".

Done (line 39-40).

> Line 36: Although the Super et al. (2018) SST reconstruction is incredibly useful, I would not consider the Miocene a "resolved" issue. The Miocene CO2 is a highly debated topic at the moment.

Revised (lines 42-3).

> Line 46 (and throughout): Add "values" after all _13C

Done (throughout).

> Line 118-121: Remove "such as" throughout. Here, the author includes every single site, so there's no need to express them as examples.

Done (lines 162-167 and throughout).

> Line 143-150: Please break down into several sentences, quite difficult to read.

Revised (lines 190-199).

> Line 175: Remove "even"

Done (line 233).

> Line 196: "Recent" what?

Revised (line 260)

Anonymous Reviewer #2

Thank you to the reviewer for their kind words about the manuscript and the constructive comments.

> As mentioned above, the main issue I have is the statement that the activation of CCMs is the sole explanation between the offset in CO2 reconstructions between the alkenone proxy and ice core record. There is no evidence that since the development of CCMs in haptophytes (which may have occurred during the late Miocene – early Pliocene) these CCMs could be turned on and off. On the contrary, a study from Van de Waal et al. 2019 (L&O Letters) suggests that haptophyte CCMs (measured in present day haptophytes) are not so adjustable even in high CO2 environments. Although this is from present day haptophytes, no mention of such findings is made, even though the data presented here is closer in age to the present-day haptophytes than those from late Miocene.

> The main conclusion of the paper also revolves around CCMs, but this topic is hardly introduced or properly explained in the introduction and explored in the discussion. This can certainly be improved. CCMs also comprise various mechanisms of acquiring C and it can be explored how alterations in these strategies may compromise alkenones being a reliable proxy for atmospheric CO2. It may have something to do with increased uptake of HCO3 relative to CO2, but no mention of this is made. Just stating CCMs are turned on or off is a bit oversimplified, especially since there is not a lot of evidence for this.

> A bit more emphasis on the comparison of the alkenone proxy to the ice core record may be made in the title and in the abstract, as it is very nice that data from multiple sites are combined and also demonstrates the pitfalls of this proxy.

I now include a new section in the Introduction outlining what is understood about CCMs in coccolithophores (new section 1.1, lines 45 -79) and have revised the Discussion (throughout the Discussion, lines 235-265), and now include more specifics of the comparison in the abstract (lines 7-8). As stated in the discussion phase, the work of Van de Waal is intriguing, although is based on a modern haptophyte evolved within the low $CO_2$ world of the Pleistocene. I have revised the language throughout to note that it is if CCMs dominate that the proxy seems to fail, and note the evidence (reviewed by Reinfelder 2011 and discussed in the new section 1.1) that coccolithophores likely supplement passive diffusion with CCMs but that the evidence is that coccolithophores on the whole do so much less effectively than some other algae.

> There are still a few mistakes and a few awkward sentences in the text.

These have now hopefully all been revised, along with all the minor revisions and corrections suggested (see marked up latexdiff version)

> Line 17-21 Quite a long and confusing sentence with conjugations that do not fit.

Revised and split (lines 19-24).

> Line 61, but also Line 84 here you state that additional corrections from the original records were removed, but you accounted for that in the fractionation with the "b" term,

right? How exactly is this term calculated for all the sites?

This is detailed in lines 128-133.

> Line 175 what do you mean here? The study you did or the one from Laws? Not clear from sentence structure, although I assume you mean your study as you refer to alkenones. If so, I would not state it like this, as you only look at sedimentary records which is not clear behavior of activation of CCMs.

This passage has now been revised (lines 232-237).

> Line 182 not sure if this is necessarily a CCM threshold or a switch maybe from one of mechanisms of the CCM (for instance a switch from $CO_2$ to $HCO_3$ uptake)

Revised (line 244-7).

> Line 187 maybe also state how SST influences aqueous $CO_2$, as this is not yet mentioned.

Revised (line 250).